# Progress and inequality in child immunization in 38 African countries, 2000–2030: A spatio-temporal Bayesian analysis at national and sub-national levels

Phuong The Nguyen[1,2]*, Ryota Nakamura[3], Hideyasu Shimadzu[4], Aminu Kende Abubakar[2,5], Phuong Mai Le[6], Huy Van Nguyen[7,8], Hoa L. Nguyen[8], Motohiro Sato[1], Ayako Honda[1], Stuart Gilmour[5]

1 Research Center for Health Policy and Economics, Hitotsubashi Institute for Advanced Study (HIAS), Hitotsubashi University, Tokyo, Japan, 2 Division of Population Data Science, National Cancer Center Institute for Cancer Control, Tokyo, Japan, 3 Department of Global Health and Development, London School of Hygiene & Tropical Medicine, London, United Kingdom, 4 Department of Data Science, Kitasato University, Kanagawa, Japan, 5 Graduate School of Public Health, St. Luke's International University, Tokyo, Japan, 6 Department of International Trials, Center for Clinical Sciences, National Center for Global Health and Medicine, Tokyo, Japan, 7 Health Innovation and Transformation Centre, Federation University, Ballarat, Victoria, Australia, 8 Department of Population and Quantitative Health Sciences, UMass Chan Medical School, Worcester, Massachusetts, United States of America

* nguyenthephuong.hmu@gmail.com, phuong.nguyen@r.hit-u.ac.jp

## Abstract

### Background

Monitoring progress and inequality in childhood immunization coverage at both national and sub-national levels is essential for refining equity-oriented health programs and ensuring equitable access to care towards achieving global targets in African countries.

### Methods and Findings

Using approximately 1 million records from 104 nationally representative Demographic and Health Surveys (DHS) conducted in 38 African countries (2000–2019), we estimated childhood immunization coverage for key indicators (BCG, MCV1, DPT3, Polio3, and Full immunization), stratified by socioeconomic status. Variations of Bayesian spatio-temporal analysis using Besag, Besag–York–Mollié (BYM) and BYM2 models were employed to assess and project the trends from 2000 to 2030. We evaluated the probability of achieving Universal Health Coverage (80% coverage) and Immunization Agenda (90% coverage) by 2030, at national and sub-national levels. Finally, we conducted a comprehensive inequality analysis using the Slope Index of Inequality (SII) and Relative Index of Inequality (RII) to assess changes over the study period.

**Data availability statement:** The data used in this study are publicly available through the Integrated Public Use Microdata Series Demographic and Health Surveys (IPUMS DHS) database. Access to the harmonized datasets requires registration and approval, and can be obtained from https://globalhealth.ipums.org/. All R code used for data analysis and modeling is available on GitHub: https://github.com/nguyenthephuong1/child-immunization-africa.

**Funding:** This work was supported by the Japan Society for the Promotion of Science (Grant Numbers 22J13600 and 22KJ2761 to PN; Grant Number 23H00049 to MS, RN, and PN). The funders had no role in the study design, data collection, analysis, decision to publish, or manuscript preparation.

**Competing interests:** The authors have declared that no competing interests exist.

**Abbreviations :** AARC, Average Annual Rate of Change; AD, Absolute Difference; BCG, Bacillus Calmette–Guérin; BYM, Besag–York–Mollié; CrI, Credible Intervals; DHS, Demographic and Health Surveys; DIC, Deviance Information Criterion; DPT, Diphtheria–Tetanus–Pertussis; INLA, Integrated Nested Laplace Approximation; IPUMS, Integrated Public Use Microdata Series; LMICs, Low- and Middle-Income Countries; MCMC, Markov chain Monte Carlo; ME, Mean Error; RII, Relative Index of Inequality; SDGs, Sustainable Development Goals; SDI, Socio-Demographic Index; SII, Slope Index of Inequality; UHC, Universal Health Coverage; WAIC, Watanabe-Akaike Information Criterion; WUENIC, WHO/UNICEF Estimates of National Immunization Coverage.

Childhood immunization coverage improved significantly across most African countries from 2000 to 2019. However, projections suggest that 12 countries are unlikely to achieve global targets for full immunization by 2030 at the national level if current trends continue. Notably, high-Socio-Demographic Index (SDI) countries such as South Africa, Egypt, and Congo Brazzaville are projected to miss immunization targets across all sub-national regions. While socioeconomic inequalities were widespread in 2000, they are projected to decline or stabilize in 36 countries by 2030, with Eswatini, Morocco, Rwanda, and Burkina Faso expected to eliminate disparities. In contrast, Nigeria and Angola are projected to face increasing inequalities or persistent large gaps. Regional disparities in both coverage and inequality remain pronounced, particularly in Central and Western Africa, where coverage remains low and inequality remains high despite overall national-level improvements. The analysis was limited to DHS surveys 2000–2019, excluding more recent data during the COVID-19 period and potentially overestimating trends in data-sparse settings.

## Conclusions

This study highlights both progress and persistent challenges in childhood immunization coverage, along with inequalities across 38 African countries. Persistent regional disparities and socioeconomic inequalities require multifaceted strategies that account for demographic, geographic, economic, and political factors to ensure equitable immunization. Greater efforts are needed to close these gaps and support global health goals for the African nations.

## Author summary
### Why was this study done?

- Vaccines are one of the most effective ways to protect children from deadly diseases, yet immunization coverage is still suboptimal in many African countries.

- Monitoring progress toward global immunization targets and reducing within-country inequalities has been challenging in many low- and middle-income countries due to limited and infrequent data, especially at sub-national levels.

- This study was conducted to generate consistent, comparable estimates of childhood immunization coverage and inequalities at sub-national levels across 38 African countries.

### What did the researchers do and find?

- We analyzed data from over 1 million households across 104 national surveys conducted between 2000 and 2019.

- Using spatio-temporal Bayesian modeling techniques, we estimated immunization coverage trends through 2030 and assessed disparities across geographic areas and socioeconomic groups.

- Although immunization coverage has improved in many settings, substantial gaps persist both within and between countries, and only a few are on track to meet global immunization targets.

## What do these findings mean?

- The results serve as a reference point for monitoring future progress and for identifying vulnerable populations at risk of being left behind.

- Policymakers and health programs must adopt more focused, equity-driven strategies to ensure all children have access to life-saving vaccines.

- As projections are based on data collected before the COVID-19 pandemic, they may not fully reflect recent disruptions and should be interpreted with appropriate caution.

## Introduction

Childhood immunization is one of the most cost-effective public health interventions, preventing millions of deaths annually and substantially reducing the burden of vaccine-preventable diseases. Since the launch of the Expanded Programme on Immunization (EPI) by the World Health Organization (WHO) in 1974, substantial progress has been made globally, particularly in low- and middle-income countries (LMICs) [1]. Specifically, vaccination programs have been estimated to have saved 153.8 million lives globally and 52.8 million lives in Africa over the past 50 years [2]. The vaccines for Bacillus Calmette–Guérin (BCG), MCV1, Polio, and Diphtheria–Tetanus–Pertussis (DPT) contribute to over 95% of lives saved through immunization [3,4]. Despite these advancements, immunization coverage remains lowest in Africa, with persistent disparities between and within African countries [5]. This situation is hindered further by the impact of the COVID-19 pandemic, logistical issues, the rising number of children in fragile and conflict zones, underscoring the need for focused monitoring and evaluation.

Global initiatives focusing on increasing immunization coverage and ensuring equitable access to vaccines have been established to advance childhood immunization [6]. The World Health Assembly's Immunization Agenda 2030 (IA2030) reflects a unified global commitment to ensuring that all children, regardless of socioeconomic status or geographic location, benefit from immunization [7]. Expanding coverage of essential health services, including immunization through equity-focused health programs, is also a central objective of key global targets, including Universal Health Coverage (UHC) and the Sustainable Development Goals (SDGs) [8]. While these efforts have gained momentum worldwide, recent trends indicate stagnation or slow progress in many resource-constrained settings, such as in African countries, raising concerns about whether global immunization targets will be met [9]. Currently, immunization coverage in Africa stands at 78% for BCG, 68% for MCV1, 70% for Polio (third dose, Polio3), and 71% for DPT (third dose, DPT3), falling well below global averages and considerably short of key global targets [10]. Persistent disparities both between and within African countries, compounded by geographic and socioeconomic inequalities, further complicate progress toward universal immunization coverage and health equity [11–13].

To address these challenges, monitoring childhood immunization coverage at both national and sub-national levels is essential for refining equity-focused health programs tailored to each country's specific needs and ensuring equitable access to care [14]. While childhood immunization coverage has been regularly monitored by WHO and UNICEF, there are persistent knowledge gaps in tracking sub-national progress and stratified analyses [7,15]. Even if a nation is on track to achieve its national immunization targets, childhood disease may persist in sub-national areas with low vaccination

rates, hence, health inequalities may remain unaddressed despite overall good progress. Moreover, sub-national analyses of long-term immunization trends, projections toward global targets, and socioeconomic-related inequalities are critical to generating granular insights beyond national statistics and routine monitoring reports. This evidence-based information is essential for informing policy decisions, optimizing resource allocation, and guiding targeted interventions to promote equitable vaccine access, advance social justice, and achieve global health goals.

While valuable, previous single-country studies on childhood immunization in Africa have often faced data limitations, such as insufficient time-series data for trend analysis and small sample sizes for stratified analyses [16]. These constraints made it challenging to assess long-term trends, generate reliable estimates at the sub-national level, and identify disparities within countries. Meanwhile, existing multi-national studies have primarily focused on national-level estimates, often overlooking sub-national variations and disparities critical for understanding localized barriers to immunization [11,12]. Notably, prior multi-country modeling efforts, such as those by Mosser and colleagues [17] and Utazi and colleagues [18], have mapped fine-scale immunization coverage across Africa, providing valuable spatial insights, but they did not incorporate socioeconomic stratifications, forward projections, or detailed inequality analyses at sub-national levels. A multi-national study with sub-national level analysis is needed to overcome these limitations by integrating both national and sub-national perspectives, enhancing statistical power, and enabling more robust trend estimations and inequality assessments across sub-national regions and socioeconomic groups.

The objectives of this study were 2-fold: (1) to estimate trends and projections of childhood immunization coverage from 2000 to 2030, and assess the probability of achieving global immunization targets at both national and sub-national levels, and (2) to examine socioeconomic-related inequalities in immunization coverage, including estimates, patterns, and trends of inequalities. Using advanced Bayesian spatio-temporal models, we leverage spatial borrowing and trend smoothing for more reliable projections across 38 African countries, even in data-sparse sub-national regions. This approach provides a more comprehensive analysis of immunization coverage, capturing long-term trends and localized disparities often missed in national-level studies, supporting more effective tracking of progress towards global immunization targets.

## Methods

### Data sources

In this secondary analysis, we collected a comprehensive set of Demographic and Health Surveys (DHS) of African countries from the Integrated Public Use Microdata Series (IPUMS) [19]. The DHS used standardized questionnaires and employed a standardized sampling design and method across all countries, typically following a stratified two-stage cluster approach, to ensure data comparability. The surveys primarily focused on women of productive age (aged 15–49 years) and their children to collect nationally representative data on population health and demographic indicators in LMICs, including Reproductive, Maternal, Newborn, and Child Health. Detailed descriptions of the sampling methodology, questionnaires, and indicator measurements of DHS have been published elsewhere [20]. We included all available DHS data for African countries from 2000 to 2019, ultimately including 1,000,435 records of living children from 104 datasets across 38 African countries. While more recent DHS rounds exist, they were neither publicly released nor integrated into the IPUMS system at the time of analysis, making systematic inclusion impractical. Furthermore, direct access to DHS datasets has been constrained following the program's suspension under the recent U.S. administration. To ensure consistency and comparability across countries, we therefore focused our analysis on the 2000–2019 period. Details of included surveys by country are provided in Table A in S1 File.

We analyzed data at sub-national regions—the first administrative-level units (Admin-1), which served as the primary unit of analysis. These units were matched to national boundaries using shape files from the DHS Spatial Data Repository. For countries with multiple survey rounds, we harmonized sub-national boundaries to align with the most recent administrative definitions. In cases where boundaries had changed over time, smaller units were merged into broader

regions to ensure consistency in regional comparisons, following established practices in spatio-temporal mapping of IPUMS DHS data [21].

## Childhood immunization

The DHS program monitored childhood immunization status through two sources: (1) from vaccination record cards provided by mothers/caretakers (primarily), and (2) verbal reports if cards were unavailable (supplemented) [20]. Consistent with the UHC and WHO global monitoring frameworks, we selected vaccination indicators tailored to African countries [7,15,22,23]. The recommended vaccination schedule includes BCG administered at birth, three doses of oral polio vaccine or inactivated polio vaccine with an optional dose at birth and mandatory doses at 6, 10, and 14 weeks, three doses of DPT at 6, 10, and 14 weeks, and one dose of MCV1 vaccine at 9 months [20]. Accordingly, we analyzed four key immunization indicators: BCG, three doses of Polio (Polio3), three doses of DPT (DPT3), and MCV1 vaccines. The primary outcome of this study was full immunization coverage, defined as the proportion of living children within a specified age group who received all recommended vaccines. In this study, we measured the main outcomes for children aged 12–23 months, a time frame widely used in previous research [7,20,24]. Information regarding estimated indicators, including definitions, denominators, and numerators, is outlined in Table B in S1 File.

We estimated childhood immunization coverage indicators for each country and year, stratified by sub-national levels and socioeconomic groups, including household wealth quintiles and maternal education levels. Wealth quintiles were derived using principal component analysis (PCA), which generated a wealth score based on household assets and services, ranking households into quintiles from poorest to richest, as per DHS survey standards. Maternal education was categorized as a binary variable, distinguishing between less educated (primary school or lower) and most educated (secondary school or higher) groups. We adjusted all estimated coverage values for the complex survey design, incorporating sampling weights, clustering, and stratification [25].

## Statistical analysis

To estimate sub-national trends in immunization coverage, we applied Bayesian spatio-temporal models incorporating spatial dependencies and temporal smoothing. In countries with only one available survey, temporal trends were inferred using the global time trend and spatial borrowing from neighboring countries and sub-national regions, with increased uncertainty reflected in the posterior estimates. The models used include the Besag model [26], Besag–York–Mollié (BYM) model [27], and enhanced BYM2 model with tailored parametrization and meaningful assignment of Penalized Complexity priors [28]. To model time trends, we applied a linear specification on the logit scale for each sub-national region, enabling projections to 2030 [29]. Alternative approaches, including autoregressive and random walk models, were tested but resulted in unstable or discontinuous trends, limiting interpretability. We also explored the use of country-level covariates (e.g., SDI, HDI), but this approach would require projecting those covariates to 2030, introducing further uncertainty. The final logit-linear model offered a balance of simplicity, interpretability, and performance across all indicators and settings, consistent with prior forecasting studies [11,30]. We conducted stratified analyses by wealth quintile and education level, incorporating random slopes by sub-national region and interaction terms with time. This allowed us to estimate subgroup-specific trajectories and project immunization coverage, AARC, SII, and RII values to 2030, assuming observed trends from 2000–2020 remain constant in the absence of major interventions. Full model details are available in the Supplementary Modeling section.

We applied Integrated Nested Laplace Approximation (INLA), a computational alternative to Markov chain Monte Carlo (MCMC) methods for conducting approximate Bayesian inference within latent Gaussian models [31]. INLA was selected over MCMC methods for its computational efficiency and ability to conduct large-scale spatio-temporal models, such as estimating immunization coverages across multiple countries and sub-national regions. Model comparison and selection were conducted using data from 2000 to 2016, with 2017–2019 held out for out-of-sample validation. We assessed model

performance using Deviance Information Criterion (DIC), Watanabe-Akaike Information Criterion (WAIC), Absolute Difference, Mean Error, and computational efficiency (runtime in seconds), evaluating each model on in-sample (2000–2016), out-of-sample (2017–2019), and full-period (2000–2019) performance. Among the six model variations, we selected **model_bym2_2**, a hierarchical BYM2 model that captures within- and between-region spatial dependencies and accounts for national-level structure. It demonstrated the best overall performance across all immunization indicators based on DIC, WAIC, average absolute difference, and mean error. Its nested spatial structure also enhanced interpretability, aligning well with the hierarchical country–region relationships in the data. Details of model comparison and validation are presented in S1 File (Table C and Figs A1–A14 in S1 File).

Under the Bayesian framework, we estimated the probability of achieving global targets for childhood immunization at national and sub-national levels. Specifically, we evaluated the UHC target (80% coverage of essential healthcare services) and the IA2030 target (90% coverage of essential childhood and adolescence vaccines) by 2030 [7,9]. To do so, we computed exceedance probabilities, which represent coverage of childhood immunization at location $i$, and is the target threshold (e.g., 0.8 for UHC, 0.9 for IA) [32]. We extracted the posterior predictive distribution and calculated the average annual rate of change (AARC) and its 95% Credible Intervals (CrI) using the formula:

$$AARC = 100 * \left[ \left( \frac{P_n}{P_0} \right)^{\frac{1}{N}} - 1 \right]$$

where: $P_n$ is coverage in the later time period (year 2030); $P_0$ is coverage in the earlier time period (year 2000); and $N$ is the number of years in the interval.

## Validation analysis

Model validation was conducted using the Absolute Difference (AD) and Mean Error (ME) between observed and predicted values, as shown in Tables D and E in S1 File [33,34]. The final model demonstrated strong accuracy, with an average AD of 2.9% across immunization indicators (ranging from 1.9% to 3.5%) and countries (ranging from 0.1% to 10.0%) [35]. While in-sample ME was close to zero, out-of-sample ME (holding out 2017–2019) was consistently positive (1.27% to 7.86%), suggesting a possible tendency to overestimate recent progress. We independently modeled full immunization coverage and its component vaccines (BCG, Polio3, DPT3, and MCV1), without imposing constraints that force full coverage to remain below its components. A post-modeling consistency check revealed only a few minor instances (0.8%) where full coverage slightly exceeded one or more components, with negligible effect on the overall findings (Table E in S1 File).

## Inequality analysis

To examine inequality in childhood immunization coverage, we calculated various indices using posterior predictive distribution, including the Slope Index of Inequality (SII) and Relative Index of Inequality (RII). These indices quantify the absolute difference in percentage points and relative ratio of coverages between the most advantaged and disadvantaged sub-groups [36,37]. We analyzed two dimensions of socioeconomic-related inequality, encompassing household wealth (richest versus poorest), and educational attainment (most educated versus less educated) at both national and sub-national levels. Furthermore, we assessed changes in both relative and absolute inequalities over time. For the RII, we calculated the AARC using the same approach described earlier. For the SII, which ranges from −100% to 100% and is not suitable for the AARC formula due to the possibility of negative values, we instead calculated the Absolute Annual Change (AAC) to quantify the trend over time.

## Sensitivity analysis

While children aged 12–23 months are commonly used for childhood immunization indicators, age group definitions vary, with some studies using 12–35 months and 24–35 months for immunization measures [38,39]. In addition, while

immunization among new birth cohorts is pivotal, achieving the elimination of highly contagious diseases necessitates assessing coverage among all age groups at sub-national levels. Thus, we conducted a sensitivity analysis of childhood immunization across different indicator definitions, including children aged 24–35 months, 12–35 months, 24–59 months, and 12–59 months. This analysis provides evidence of potential disparities in immunization coverage between age groups, identifies underserved child age groups, and informs the development of targeted intervention strategies [40].

Our projections assume that historical trends, observed throughout the study period, continue to the future. However, recent implementations of national policies and global initiatives may have had a greater impact on trends in recent years (e.g., 2010–2019). As a result, recent trends are likely more indicative of future projections [41]. To address this, we conducted a sensitivity analysis comparing the 2030 projections using two data periods: one covering the entire period from 2000 to 2019 and the other using only recent data from 2010 to 2019. In addition, given the variation in progress across sub-national regions within countries, we compared the national 2030 projections with an alternative scenario based on the observed AARC of its best-performing region, defined as the region with the highest AARC during the 2000–2019 period. Finally, given the comparable performance of the three models, we compared their 2030 projections as a sensitivity analysis of model selection, using the Besag, BYM, and BYM2 models.

All statistical analyses were performed in R (version 4.3.1). This study is reported as per RECORD guideline (S1 Checklist).

### Ethics statement

This study is a secondary analysis of publicly available, de-identified data with no identifiable personal information. As such, no ethical approval was required.

### Inclusivity in global research

Additional information regarding the ethical, cultural, and scientific considerations specific to inclusivity in global research is included in the Supporting Information (S2 Checklist).

## Results

### Childhood immunization coverage at the national level, 2000–2030

Fig 1 shows the coverage of full immunization and the four key vaccines—BCG, MCV1, Polio3, and DPT3—for 2000 and 2020, with projections for 2030. It also presents the AARC over 2000–2030 and the proportion of regions expected to achieve the UHC (80% coverage) and IA (90% coverage) targets by 2030. Immunization coverage improved significantly in most African countries, with the majority showing statistically significant positive AARC. However, declining trends were observed in Egypt (MCV1 and Full immunization), Congo Brazzaville (Polio3, DPT3, and Full immunization), and Guinea (all five indicators). If current trends continue, most countries are projected to meet or exceed both UHC (80%) and IA2030 (90%) targets for BCG, MCV1, Polio3, and DPT3 by 2030, reaching UHC in 33–37 countries and IA targets in 30–33 countries. In contrast, only 26 of 38 countries are projected to achieve the UHC target for full immunization, and just 19 are expected to meet the IA2030 target. At the sub-national level, only Eswatini, Morocco, Kenya, Rwanda, Malawi, Burkina Faso, and Niger are projected to achieve the UHC target across all sub-national regions, with only Malawi and Burkina Faso reaching IA2030 target across all regions. Conversely, Comoros, Angola, and Guinea are not projected to meet either target in any region. Coverage levels and the probabilities of reaching the global immunization targets by 2030, both nationally and stratified by wealth quintile and educational level, are presented in Tables F–H in S1 File. Tables I–K provide information on the observed and projected AARC over 2000–2020 and 2020–2030, along with the additional AARC values needed for countries to meet the 2030 targets, at the national level and stratified by wealth quintile and educational level.

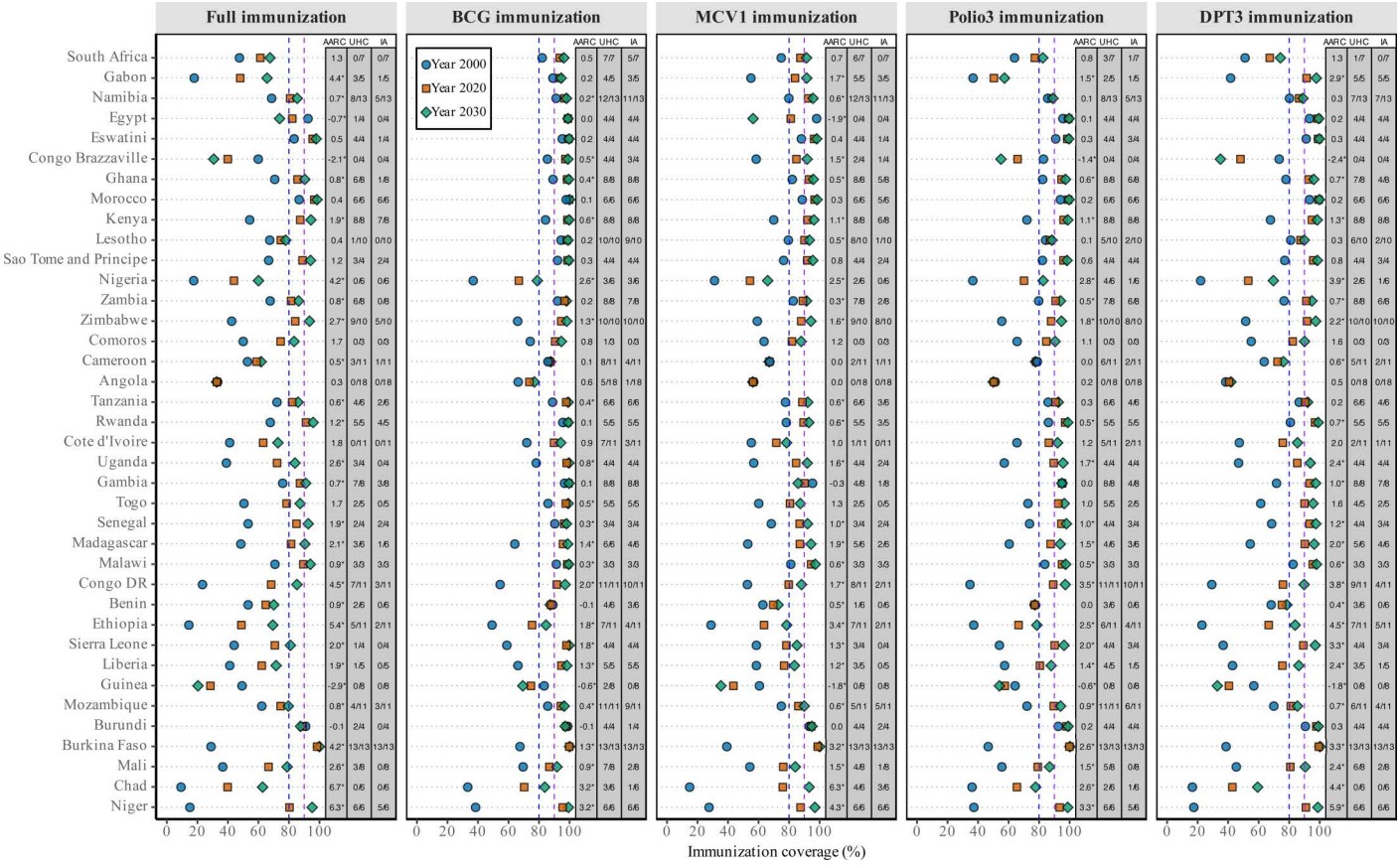

**Fig 1. Childhood immunization coverage in 2000, 2020, and 2030, average annual rate of change (AARC) over 2020–2030, and proportion of regions expected to reach 2030 targets across 38 African countries.** Countries (Y-axis) are ordered by Socio-Demographic Index (SDI), from highest (top) to lowest (bottom). Vertical dashed lines mark global targets: the blue line indicates the UHC benchmark (80% coverage), and the purple line indicates the IA2030 target (90% coverage). Abbreviations: AARC, Average Annual Rate of Change; UHC, Number of regions per country projected to reach 80% coverage by 2030; IA, Number of regions projected to reach 90% coverage by 2030.

## Childhood immunization coverages at the sub-national level, 2000–2030

Fig 2 presents childhood immunization coverage in 2000, 2020, and 2030 at the regional level in 38 African countries. In 2000, full immunization coverage was generally low across Africa, except in parts of Northern (Egypt, Morocco) and Southern Africa (Namibia, Tanzania, Mozambique). Significant regional disparities were evident, with the lowest coverage levels. often below 40%, observed in Central and Western Africa, while Eastern and Southern Africa showed higher coverage levels, typically above 60%. This pattern of disparity was consistent across all immunization indicators, although BCG generally showed higher coverage (approximately 10% higher) than MCV1, DPT3 and Polio3, as reflected by the greater presence of blue areas in the maps. By 2030, immunization coverage is projected to improve significantly, with most sub-national regions expected to surpass the 80% target if trends continue, although some areas in Central and Western Africa are predicted to lag behind. Disparities persisted, with Eastern and Southern Africa maintaining greater progress than Central and Western Africa.

## Socioeconomic-related inequality at the national level

Fig 3 shows estimated full immunization coverage by wealth quintile and educational level in 2000 and 2020, along with projections for 2030. It also illustrates socioeconomic inequalities in coverage, measured by the SII and RII. Changes in

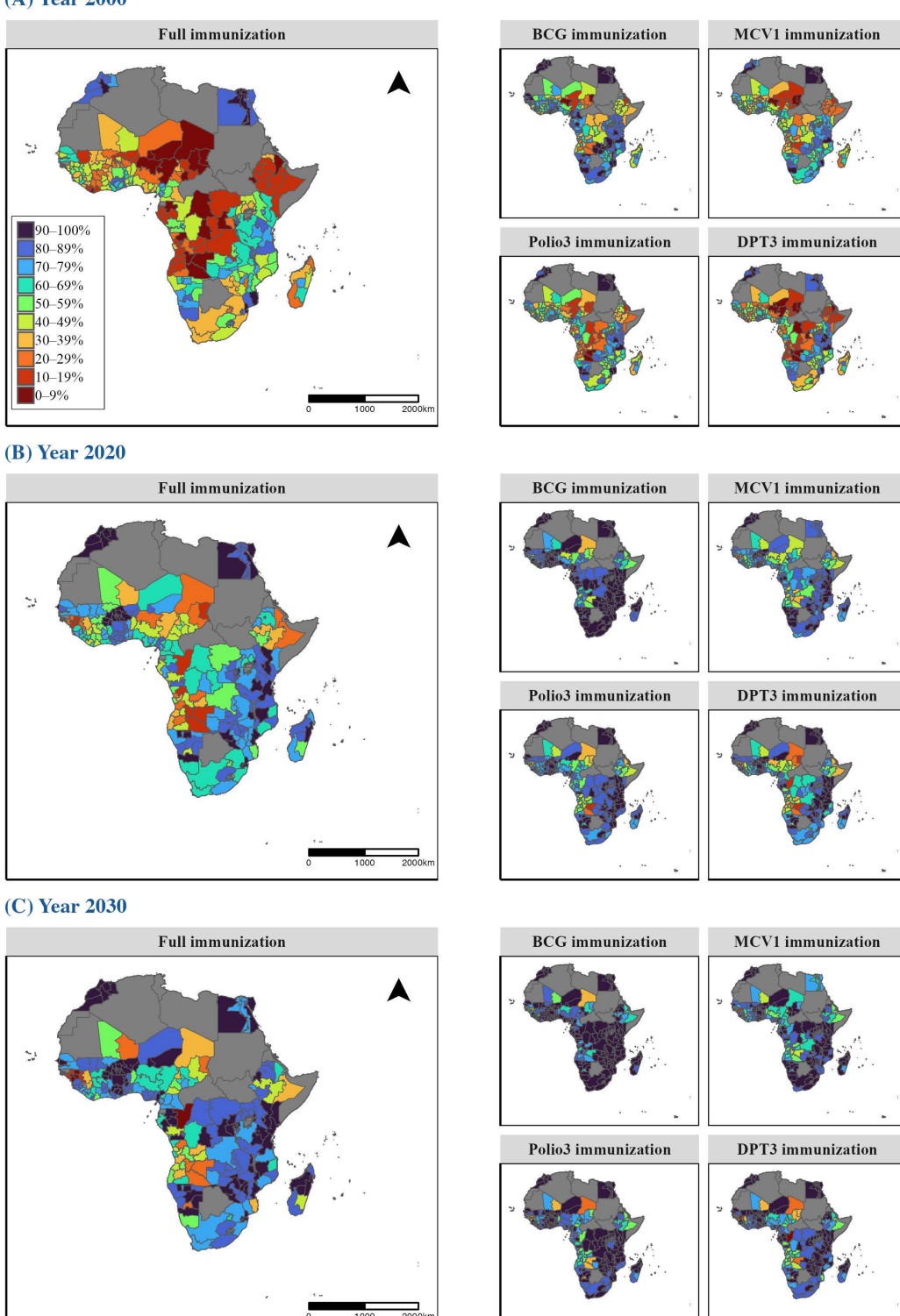

**Fig 2. Maps of childhood immunization coverage in African countries at regional level for (A) 2000, (B) 2020, and (C) 2030.** Coverage levels are displayed in 10 color-coded categories, ranging from lowest (0%–9%) to highest (90%–100%). Countries shaded in gray indicate missing or unavailable data. Basemap from GADM (https://gadm.org/).

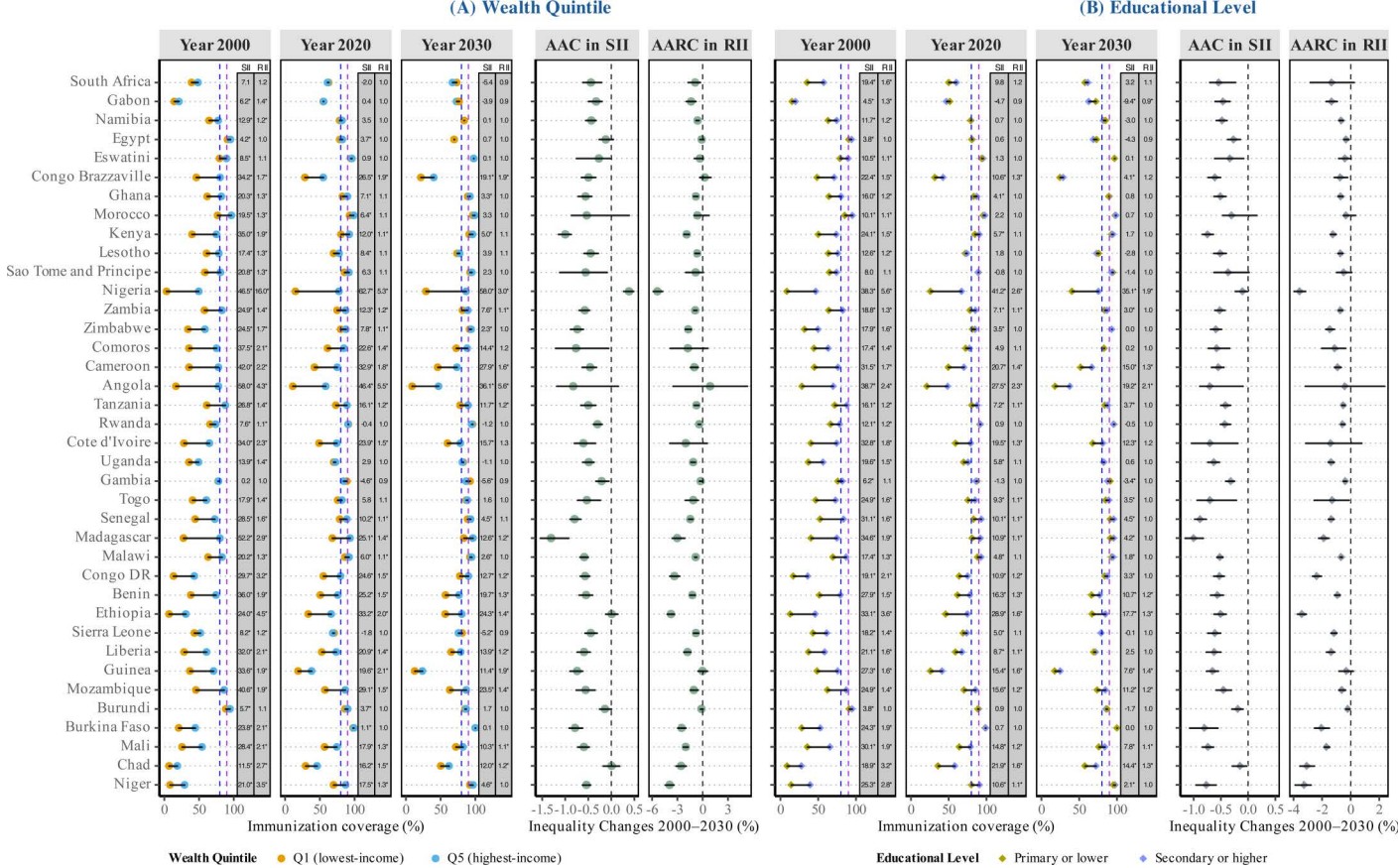

**Fig 3. Full immunization coverage in 2000, 2020, and 2030 by wealth quintile and educational level, with socioeconomic-related inequality indices (SII, RII) and changes over 2000–2030 (AARC).** The Y-axis lists 38 African countries, ordered by Socio-Demographic Index (SDI) from highest (top) to lowest (bottom). Vertical dashed lines mark global targets: the blue line indicates the UHC benchmark (80% coverage), and the purple line indicates the IA2030 target (90% coverage). Abbreviations: SII, Slope Index of Inequality; RII, Relative Index of Inequality; AARC, Average Annual Rate of Change (2000–2030).

absolute and relative inequalities from 2000 to 2030 are represented by AARC of SII and RII. In 2000, the highest-income and most-educated groups had substantially higher immunization coverage than the lowest-income and least-educated groups. The gaps reached a maximum of 58% in absolute terms or 4-fold in relative terms for wealth quintiles, with a similar but smaller disparity observed for educational levels. Socioeconomic inequalities in childhood immunization coverage were significant across all African countries. By 2030, most countries are projected to reduce these gaps, with declines in SII and RII expected under current trends. Specifically, 36 out of 38 countries are projected to show either significantly negative AAC or AARC values, indicating declining inequality trends, or in the case of Eswatini, Morocco, Rwanda, and Burkina Faso, a potential elimination of socioeconomic-related inequalities in childhood immunization coverage if current progress continues. Conversely, Nigeria exhibited increasing wealth-related inequality, with a statistically significant positive AAC in SII, while Angola maintained large and persistent gaps between socioeconomic groups over the study period. Immunization coverages by wealth quintile and education level for BCG, MCV1, DPT3, and Polio3 are shown in S1 File (Figs A15–S18). Detailed estimates and changes in socioeconomic-related inequalities are provided in S1 File (Tables L and M in S1 File).

## Socioeconomic-related inequality at the sub-national level

Fig 4 shows maps of socioeconomic inequality in full immunization coverage, represented by the SII, at sub-national level for 2000, 2020, and projections for 2030. In 2000, sizeable wealth-related inequalities were observed across most African areas, indicating substantial gaps in immunization coverage between the wealthiest and poorest groups. Central and Western Africa exhibited the most pronounced inequalities, with many areas showing SII values exceeding 30%, highlighting heterogeneity in the degree of socioeconomic-related inequalities at the sub-national level. By 2030, wealth-related inequalities are projected to decline in many sub-national regions. However, sub-national disparities are projected to persist, with higher socioeconomic-related inequalities in Central and Western Africa compared to other areas. Education-related inequalities followed similar but less pronounced geographic patterns and temporal trends compared to wealth-related inequalities. Detailed maps of socioeconomic inequalities for BCG, MCV1, DPT3, and Polio3 are provided in S1 File (Figs A19–A22).

Fig 5 presents a two-dimensional graph showing changes in absolute (SII) and relative (RII) socioeconomic inequalities from 2000 to 2030 across all immunization indicators in 38 African countries, exhibiting broadly similar patterns for

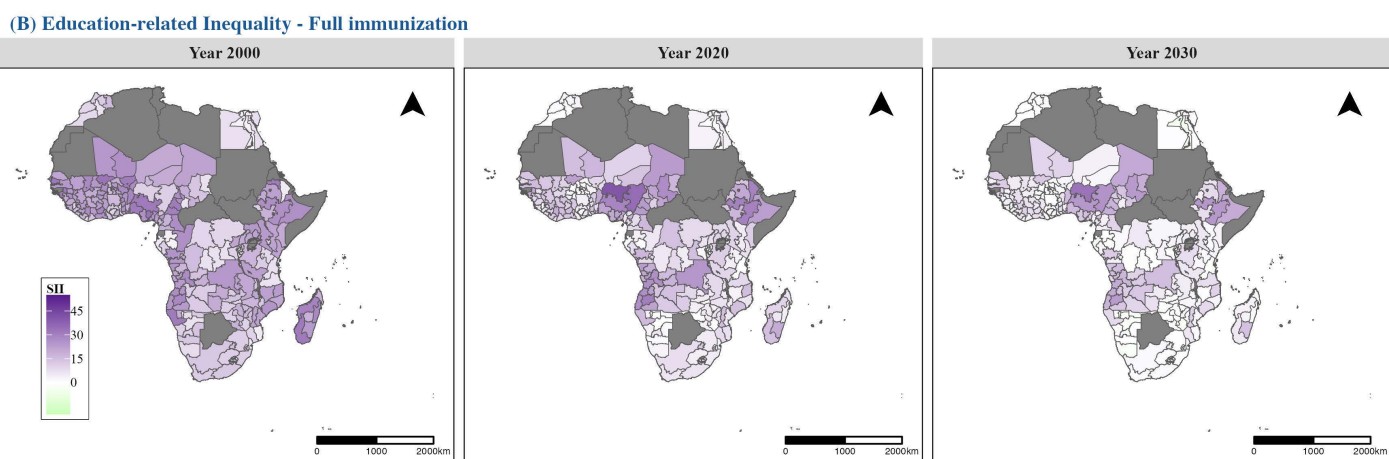

**Fig 4. Maps of socioeconomic-related inequalities in full immunization coverage at the regional level in 2000, 2020, and 2030: (A) Wealth-related inequality and (B) Education-related inequality.** Inequality is measured using the Slope Index of Inequality (SII). Countries shown in gray indicate data not available. Basemap from GADM (https://gadm.org/).

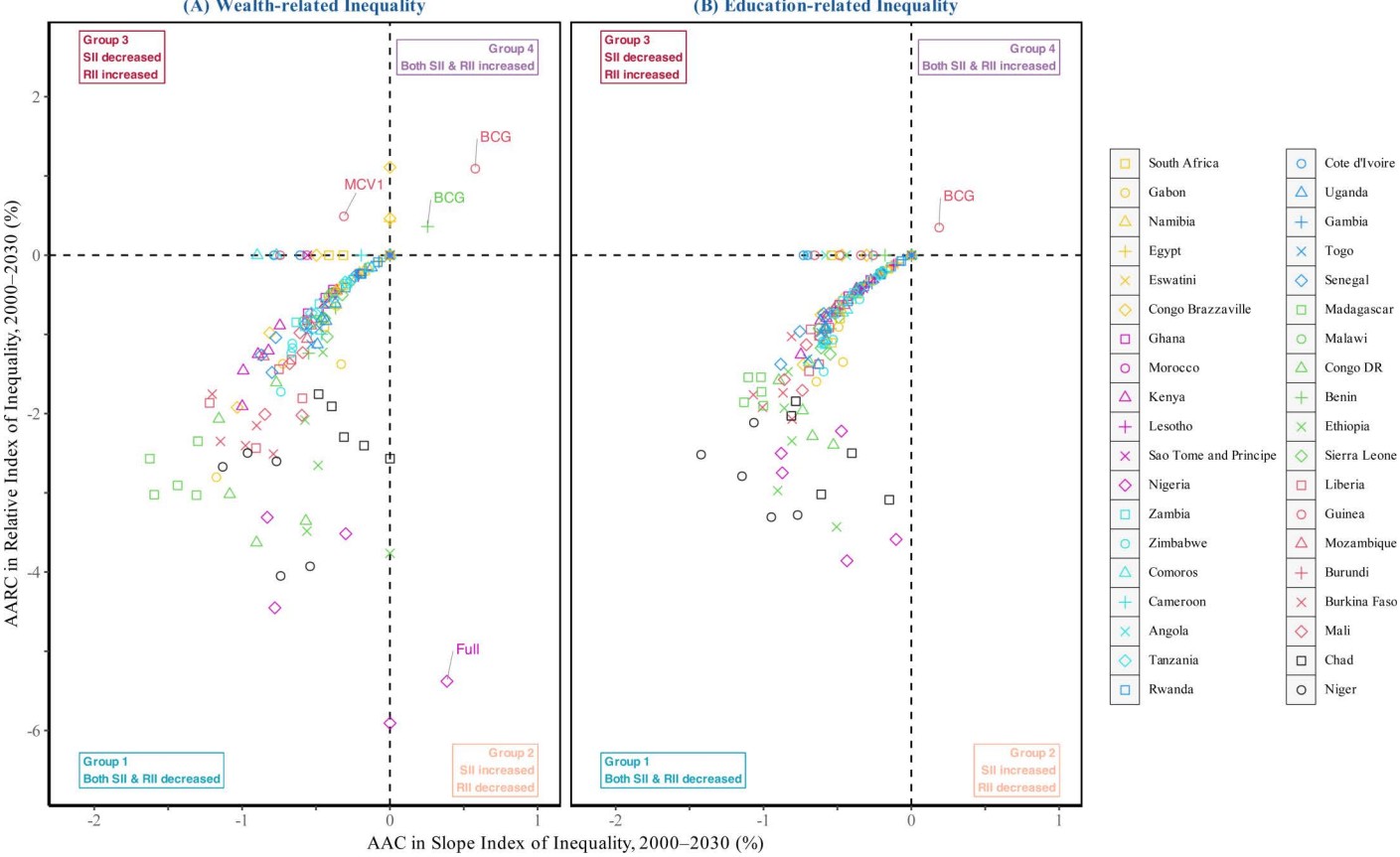

**Fig 5. Two-dimensional graph of changes in socioeconomic-related inequalities in childhood immunization coverage (2000–2030) in 38 African countries.** The analysis includes five immunization indicators: Full Immunization, BCG, MCV1, DPT3, and Polio3. Abbreviations: AARC, Average Annual Rate of Change; AAC, Absolute Annual Change; SII, Slope Index of Inequality; RII, Relative Index of Inequality.

wealth- and education-related inequalities. Countries and indicators are grouped into four categories: (1) progress in both SII and RII, indicating reductions in absolute and relative gaps; (2) progress in absolute terms but persistent relative inequality; (3) progress in relative terms but persistent absolute inequality; and (4) worsening inequalities on both scales. While increases or decreases in SII or RII can be assessed individually, interpreting their combined behavior offers deeper insights into where progress or setbacks are concentrated. For example, reductions in both SII and RII signal progress toward equity; stable SII with declining RII suggests that both disadvantaged and advantaged groups are improving, but at different rates; while rising RII typically points to stagnation or decline among disadvantaged groups. The majority of countries and indicators fell into Group 1, reflecting projected progress toward greater equality. Group 2 patterns suggest slower gains among disadvantaged groups, observed only for Nigeria's full immunization. Group 3 indicating overall declines, observed only for Guinea's MCV1 immunization, consistent with patterns observed in Figs 1 and Figs A16 (S1 File). Only two countries fell into Group 4, where both SII and RII worsened, including BCG in Benin and Guinea. These groupings help policymakers pinpoint where targeted interventions are most urgently needed to close persistent or widening gaps. Complementary analyses comparing 2000–2020 (observed) and 2020–2030 (projected) trends, presented in Figs A23–A24 in S1 File, provide additional insights into pass progress and future challenges. While six country-indicator combinations showed increasing inequality over 2000–2020, only two are projected to experience rising inequality in 2020–2030, suggesting a potential shift toward more equitable progress if current trajectories are maintained.

Figs A25–A29 in S1 File further detail sub-national regions experiencing rising inequalities across all five immunization indicators.

## Sensitivity analysis

Fig 6 presents a sensitivity analysis of selected immunization indicator definitions in Full immunization coverage. The analysis considered commonly used definitions, namely the primary measurement in this study (12–23 months), those used in previous studies (12–35 months and 24–35 months), and broader cohorts covering all children under 5 years (12–59 months and 24–59 months). The results showed nearly identical coverage levels and trends over time for narrower age groups (12–23, 12–35, and 24–35 months) across all African countries, indicating that our findings are consistent regardless of indicator definition. In contrast, the broader all-age approach yielded markedly different results in 19/38 countries, projecting much lower immunization coverage for 2030 and exhibiting declining trends from 2000 to 2030. This fact underscores disparities in immunization coverage across child cohorts in these countries. The sensitivity analysis using recent trends (2010–2019), presented in Fig A30 in S1 File, projects lower full immunization coverage by 2030 and slower progress, as indicated by smaller AARC values. Fig A31 in S1 File shows the national projections based on the

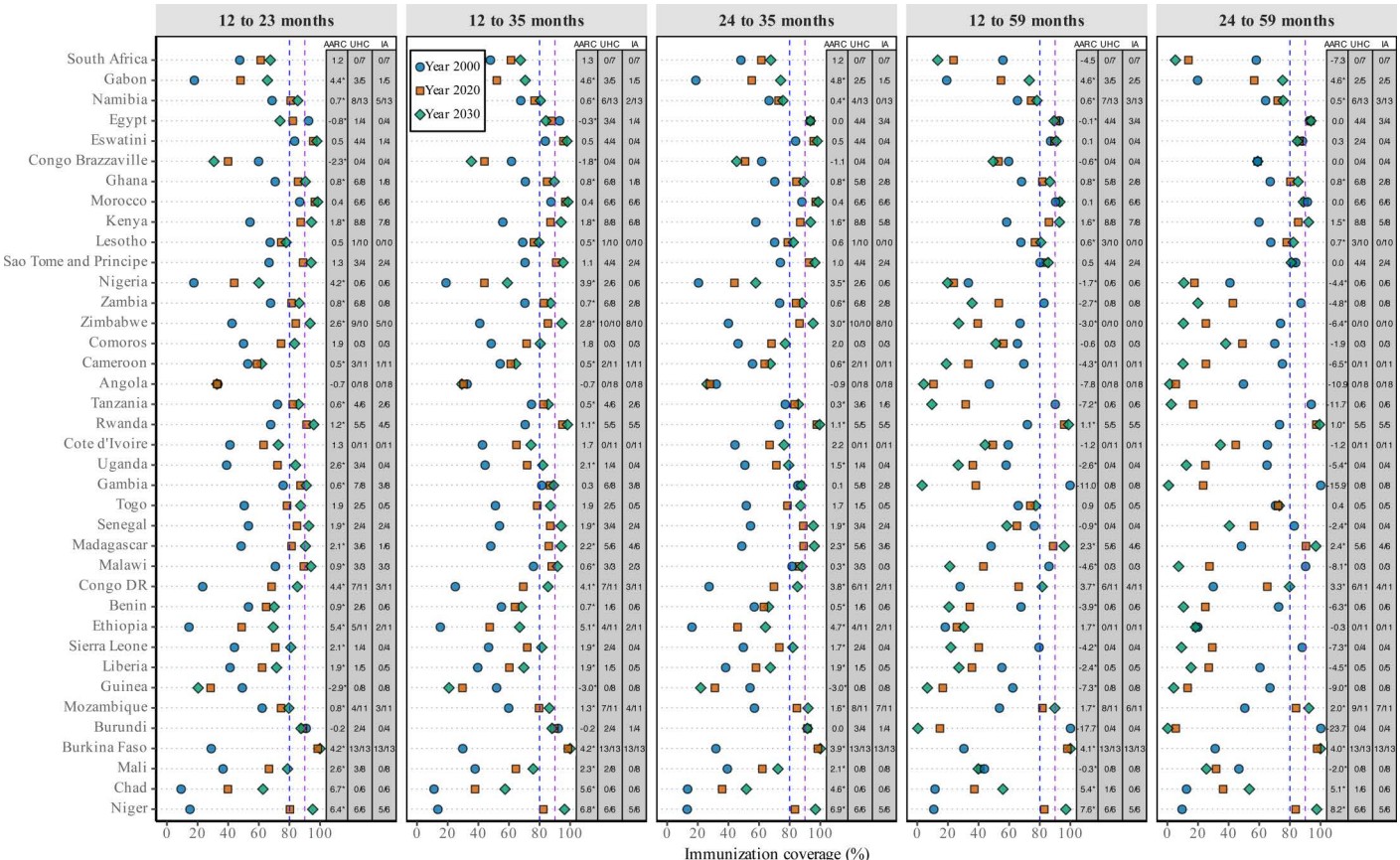

**Fig 6. Sensitivity analysis of age group selection of Full immunization indicators in coverage estimates, projections, AARC, and proportion of sub-national regions achieving 2030 targets across 38 African countries.** The Y-axis lists 38 African countries, sorted by Socio-Demographic Index (SDI) from highest (top) to lowest (bottom). Vertical dashed lines mark global targets: the blue line indicates the UHC benchmark (80% coverage), and the purple line indicates the IA2030 target (90% coverage).Abbreviations: AARC=Average Annual Rate of Change; UHC, Number of regions per country projected to reach 80% coverage by 2030; IA, Number of regions projected to reach 90% coverage by 2030.

observed AARC of best-performing region in each country. This alternative scenario illustrates the potential gains that could be achieved if all regions followed the trajectory of their top-performing one, providing valuable insight for policymakers seeking to accelerate national progress through targeted regional strategies. Fig A32 in S1 File presents a sensitivity analysis comparing projections from the Besag, BYM, and BYM2 models. BYM and BYM2 yielded similar results, while Besag differed in some countries, suggesting that including both structured and unstructured spatial effects improves model stability and projection robustness.

## Discussion

This study provides a comprehensive analysis of trends, projections, and inequalities in childhood immunization coverage across 38 African countries from 2000 to 2030 at both national and sub-national levels. Most countries showed consistent and significant improvements in coverage and are projected to meet or exceed the 2030 targets for BCG, MCV1, Polio3, and DPT3 if current trends continue, aligning with recent forecasts [42]. The findings presented are consistent with data from WHO/UNICEF Estimates of National Immunization Coverage (WUENIC), which show that many African countries have surpassed 80% coverage for childhood immunization indicators at the national level since 2020 [5]. These achievements are likely the result of sustained progress driven by decades of national and sub-national initiatives along with international support aimed at prioritizing immunization [43,44].

However, progress towards full immunization coverage remains slow in 12 African countries examined, with many sub-national regions within these nations also falling short of global targets. In most African nations, challenges related to vaccine affordability (including both direct and indirect costs), accessibility, and availability remain major obstacles, driven by weak primary healthcare systems and limited resources [45]. Vaccine stockouts are a persistent issue across sub-Saharan Africa, and are further exacerbated in resource-limited regions, which may also suffer shortages of trained personnel or unreliable electricity for cold chain management [46]. For instance, in Kenya, frequent stockouts discourage families from seeking immunization services at health facilities [47]. Socioeconomic factors remain critical barriers to the accessibility of immunization programs in Africa, influencing maternal health literacy, health-seeking behaviors, childcare practices, and access to healthcare facilities [48]. However, some low-Socio-Demographic Index (SDI) African nations have achieved relatively high immunization coverage, largely due to decades of strong and sustained support from international organizations such as WHO and UNICEF. While international funding ensures that child vaccinations are provided free of charge, which improves affordability, indirect costs such as transportation and accommodation when long-distance travel is required, continue to cause considerable financial challenges for the poorest populations when they access immunization programs [12]. In addition, the recent USAID shutdown and cuts to U.S. funding for UN organizations threaten childhood immunization progress in Africa, as these programs are key supporters of vaccine delivery, health system strengthening, and maternal and child health services [49]. The suspension of health development assistance for African countries could disrupt immunization efforts, widen regional disparities, and reverse decades of progress, making global immunization targets even harder to reach. Consequently, evidence-based cost-effective programs and sustained investments will be essential to ensuring equitable immunization progress across Africa and achieving global immunization targets [50]. Future research on the cost-effectiveness of immunization programs could help optimize resource allocation and develop sustainable financing strategies amid declining global health funding [51].

Conversely, some high-SDI African nations showed relatively low childhood full immunization coverage, with South Africa, Egypt, and Congo Brazzaville not projected to achieve global targets in all sub-national regions under current assumptions. While previous studies suggest a positive correlation between GDP per capita and vaccination coverage, our findings indicate that this relationship is inconsistent between African nations, with low immunization coverage and disparities persist despite economic growth [6,12,52]. These findings suggest that economic growth alone might be insufficient to drive immunization progress in African nations. Indeed, high-SDI African nations may still have weak immunization programs due to inefficient primary healthcare systems, misallocation of resources, or political instability [45]. Specifically,

political instability and armed conflict hinder vaccination accessibility and availability by weakening health systems and endangering the lives of health workers. In northern Nigeria, armed conflict within 10 km of a child's residence reduces the likelihood of that child receiving any vaccinations by nearly 50% [53]. In addition, fragile settings may be more sensitive to demand-side barriers such as limited awareness of vaccination by parents, misinformation, and lack of understanding of programs [13]. In particular, vaccine hesitancy, recognized as one of the top global health threats, could further reduce vaccination uptake and weaken immunization programs. In addition to internal factors such as cultural beliefs, religious opposition, mistrust in vaccines, and skepticism toward healthcare providers, vaccine hesitancy in LMICs can be influenced by external sources, including anti-vaccine movements from high-income nations and the promotion of misinformation by political figures [54]. A well-known example is Samoa's suspended vaccination program and the influence of anti-vaccine promoter Robert F. Kennedy Jr. in 2019, which led to a severe measles outbreak, resulting in thousands of infections and dozens of child fatalities [55]. This emphasizes the devastating impact of vaccine hesitancy and the need for on-going immunization programs, resilient primary healthcare systems, and greater public awareness to prevent outbreaks of infectious diseases in African nations [56].

Using spatio-temporal models, this study incorporated neighborhood effects at the sub-national level to capture variations in childhood immunization coverage in different regions. Notable sub-national disparities were observed, with some regions in Africa, particularly Central Africa, showing persistently low coverage despite improvements at the national level. Beyond that, this study also highlights the sub-national variations in socioeconomic-related inequality, with some regions exhibiting substantially higher levels of inequality than others. In 2000, pronounced socioeconomic-related inequalities in childhood immunization coverage were observed in all African countries, consistent with previous studies that found lower vaccination rates among children from less wealthy households or those with mothers of lower educational status [57]. By 2030, most of the study countries are projected to reduce socioeconomic-related inequalities, with Eswatini, Morocco, and Burkina Faso anticipated to eliminate major gaps if current progress continues. However, persistent or worsening inequalities are projected in countries such as Congo Brazzaville, Nigeria, Angola, Guinea, which have some of the highest global proportions of unvaccinated children [58]. Large and rapidly growing populations, particularly in Nigeria, place further strain on healthcare systems, making it even more challenging to achieve equitable immunization coverage [59]. Addressing these challenges requires multifaceted programs that consider demographic, geographic, socioeconomic, and political factors. Comprehensive approaches are essential to reduce inequality and ensure equitable access to immunization services.

Our sensitivity analysis using the broader all-age approach revealed remarkably lower immunization coverage for 2030 and declining trends in half of the countries, highlighting disparities among age groups whereby certain cohorts missed opportunities in these African nations. Notably, eliminating or interrupting highly transmissible infectious diseases like MCV1 requires more than achieving the UHC target (80%). It necessitates reaching the herd immunity threshold, which depends on the basic reproduction number ($R_0$) of the disease and vaccine effectiveness (VE) within a specific population, estimated at around 95% for MCV1 in African nations [60,61]. Moreover, herd immunity among new birth cohorts (children aged 12–23 months) alone is still insufficient; sustained herd immunity across all birth cohorts at the regional level is essential for the effective elimination of infectious diseases and the prevention of future outbreaks. To achieve this, efforts must focus on addressing missed opportunities and reducing cohort-based inequalities in childhood immunization coverage. This requires a comprehensive approach, including enforcing booster vaccinations, routinely monitoring population immunity to identify left-behind groups, and implementing school-based interventions, including entry immunization checks and school-based vaccination programs [62,63].

We conducted another sensitivity analysis using recent trends (2010–2019), which projected lower full immunization coverage by 2030 and slower progress, as reflected by smaller AARC values. This suggests a deceleration in coverage improvements during the recent period, consistent with global reports, including WUENIC estimates, that document plateauing immunization rates in many countries [5]. Notably, this stagnation was observed even before the COVID-19

pandemic, which further disrupted routine childhood immunization programs across Africa due to lockdowns, healthcare system strain, and interruptions in vaccine delivery [64]. These findings highlight the urgent need for additional efforts and targeted strategies to sustain and accelerate progress, as emphasized by the *Immunization Agenda 2030* [7]. At the same time, projections based on overall trends (2000–2019) yield higher coverage projections and better representation of regional variation. However, this approach may overestimate future progress by smoothing over recent stagnation, and thus findings should be interpreted with caution. On the other hand, it is crucial to assess whether immunization rates have rebounded post-pandemic and how the trends observed different with the present findings [65]. Examining regional variations in immunization recovery can also provide valuable insights into health system resilience and policy responses across different countries. This future study direction will need to leverage DHS surveys conducted during (2020–2023) and after (post-2023) the COVID-19 pandemic to evaluate recovery patterns. However, the DHS program, a key source of data for tracking immunization trends, relies on USAID funding and is currently suspended, with its future uncertain for at least the next four years [66]. This funding gap may lead to serious data shortages, making it more difficult to monitor coverage, identify disparities, and guide interventions effectively. Without sustained international support, vulnerable populations face increased risks of vaccine-preventable disease outbreaks, emphasizing the urgent need for alternative funding mechanisms to safeguard immunization efforts. To ensure continuous tracking and evaluation, local-based and sustainable monitoring systems should be developed, integrating national health surveillance programs and community-based reporting frameworks [67].

This study provides a comprehensive and policy-relevant assessment of childhood immunization coverage across 38 African countries, leveraging a large dataset of over 1 million households and advanced Bayesian spatio-temporal models. Strengths include the integration of neighborhood effects to improve sub-national estimates, stratified analyses by socioeconomic groups, quantification of inequalities, and estimation of the probability of achieving UHC and IA2030 targets at the regional level. Extensive sensitivity analyses, including comparisons across indicator definitions, time periods, and model specifications, further reinforce the robustness and generalizability of the findings. However, a few limitations warrant consideration. First, the reliance on secondary survey data may introduce biases due to recall errors or incomplete vaccination records [68]. Second, the analysis was limited to DHS surveys conducted between 2000 and 2019, based on the availability of harmonized data through IPUMS. While more recent surveys may reflect the impact of COVID-19, they were not yet publicly available or consistently integrated, and their absence remains a limitation. Nonetheless, the projections provide a useful baseline for evaluating future progress and the resilience of health systems [69]. Although we focused on DHS due to its comparability and wide coverage, other sources, such as Multiple Indicator Cluster Surveys, also offer valuable sub-national data. Yet, methodological differences in sampling and indicator definitions complicate integration, and future work could explore harmonizing these data sources [70]. While the logit-linear time trend structure allows for interpretable and consistent projections, it may over-smooth local variation in data-sparse settings or diverge from estimates like WUENIC, which incorporate survey, administrative, and country-reported data [5]. This is particularly relevant in countries with only one survey, where projections are more reliant on global trends. The consistently positive out-of-sample ME suggests a tendency to overestimate recent progress, possibly reflecting the stagnation in coverage trends, and should be interpreted with appropriate caution. Nevertheless, this method reflects common epidemiological patterns, for instance, faster initial gains and slowing progress at higher coverage levels, and has been validated in previous global health modeling work [11,30]. Finally, while this study focused on service coverage, understanding the extent to which improvements translate into better child health outcomes, such as reductions in mortality or disease burden, remains essential [71,72].

In conclusion, this study provides a comprehensive assessment of progress and inequality in childhood immunization across 38 African countries, underscoring both achievements and persistent gaps. The slower projected progress in several countries, including those with higher SDI, highlights the need for integrated efforts that strengthen primary healthcare, improve resource allocation, ensure sustainable funding, and engage communities to address both supply- and

demand-side barriers. Regional disparities in coverage and socioeconomic inequality call for targeted, context-specific strategies that consider demographic, geographic, and political realities. As health development assistance for LMICs declines, renewed commitment is essential to sustain progress and avoid setbacks; without this, global goals such as UHC and the SDGs may remain out of reach for many African countries.

## Supporting information

**S1 File.** **Table A.** Description of the included DHS surveys by countries. **Table B.** List of child immunization indicators with definitions and calculations. **Table C.** Model comparison and selection. **Table D.** Model validation by immunization indicators. **Table E.** Model validation by countries. **Table F.** Immunization coverage 2000–2030, and % reach global targets, at the national level. **Table G.** Immunization coverage 2000–2030, and % reach global targets, by wealth quintile. **Table H.** Immunization coverage 2000–2030, and % reach global targets, by educational level. **Table I.** AARC and Additional values required to reach 2030 targets, at national level. **Table J.** AARC and Additional values required to reach 2030 targets, by wealth quintile. **Table K.** AARC and Additional values required to reach 2030 targets, by educational level. **Table L.** Wealth-related inequality in childhood immunization coverage, 2000–2030. **Table M.** Education-related inequality in childhood immunization coverage, 2000–2030. **Fig A1.** Residual histogram for 6 comparing models. **Fig A2.** Predicted vs. observed values plot for 6 comparing models. **Fig A3.** Residual vs. fitted plot for 6 comparing models. **Fig A4.** Posterior Distribution of BCG immunization coverage for 6 comparing models. **Fig A5.** Trends in BCG immunization coverage at regional levels, using model_besag_1. **Fig A6.** Trends in BCG immunization coverage at regional levels, using model_besag_2. **Fig A7.** Trends in BCG immunization coverage at regional levels, using model_bym_1. **Fig A8.** Trends in BCG immunization coverage at regional levels, using model_bym_2. **Fig A9.** Trends in BCG immunization coverage at regional levels, using model_bym2_1. **Fig A10.** Trends in BCG immunization coverage at regional levels, using model_bym2_2. **Fig A11.** Trends in MCV1 immunization coverage at regional levels, using model_bym2_2. **Fig A12.** Trends in Polio3 immunization coverage at regional levels, using model_bym2_2. **Fig A13.** Trends in DPT3 immunization coverage at regional levels, using model_bym2_2. **Fig A14.** Trends in Full immunization coverage at regional levels, using model_bym2_2. **Fig A15.** BCG coverage, inequality indices (SII, RII), and changes over 2000–2030 (AARC). **Fig A16.** MCV1 coverage, inequality indices (SII, RII), and changes over 2000–2030 (AARC). **Fig A17.** Polio3 coverage, inequality indices (SII, RII), and changes over 2000–2030 (AARC). **Fig A18.** DPT3 coverage, inequality indices (SII, RII), and changes over 2000–2030 (AARC). **Fig A19.** (A) Wealth- and (B) Education-related inequality in BCG coverage. **Fig A20.** (A) Wealth- and (B) Education-related inequality in MCV1 coverage. **Fig A21.** (A) Wealth- and (B) Education-related inequality in Polio3 coverage. **Fig A22.** (A) Wealth- and (B) Education-related inequality in DPT3 coverage. **Fig A23.** Two-dimensional graph of changes in socioeconomic-related inequalities, 2000–2020. **Fig A24.** Two-dimensional graph of changes in socioeconomic-related inequalities, 2020–2030. **Fig A25.** Inequality changes in Full immunization coverage (2000–2030) with region names. **Fig A26.** Inequality changes in BCG immunization coverage (2000–2030) with region names. **Fig A27.** Inequality changes in MCV1 immunization coverage (2000–2030) with region names. **Fig A28.** Inequality changes in Polio3 immunization coverage (2000–2030) with region names. **Fig A29.** Inequality changes in DPT3 immunization coverage (2000–2030) with region names. **Fig A30.** Sensitivity analysis of using recent trend for projections of Full immunization coverage. **Fig A31.** Sensitivity analysis of using AARC of best-performing region for 2030 national projections. **Fig A32.** Sensitivity analysis of model selection for projections of Full immunization coverage.
(PDF)

**S1 Checklist.**
(PDF)

**S2 Checklist.**
(PDF)

## Acknowledgments

We would like to express our sincere appreciation to colleagues and public health experts from several African countries, including those who provided valuable feedback during the 2024 World Congress of Epidemiology in South Africa, which helped strengthen the contextual interpretation of this study. We also thank Dr. Kota Katanoda of the National Cancer Center Japan for his critical review and insightful comments.

**Declaration of Generative AI Use:** The authors independently drafted, revised, and edited the manuscript. ChatGPT was used solely for minor language refinement, with all modifications carefully reviewed and approved by the authors.

## Author contributions

**Conceptualization:** Phuong The Nguyen, Stuart Gilmour.

**Data curation:** Phuong The Nguyen, Phuong Mai Le.

**Formal analysis:** Phuong The Nguyen.

**Funding acquisition:** Phuong The Nguyen, Motohiro Sato.

**Investigation:** Phuong The Nguyen.

**Methodology:** Phuong The Nguyen, Ryota Nakamura, Hideyasu Shimadzu, Stuart Gilmour.

**Project administration:** Phuong The Nguyen, Phuong Mai Le.

**Resources:** Phuong The Nguyen, Motohiro Sato.

**Software:** Phuong The Nguyen.

**Supervision:** Phuong The Nguyen, Stuart Gilmour.

**Validation:** Phuong The Nguyen, Ryota Nakamura, Hideyasu Shimadzu, Huy Van Nguyen, Hoa L. Nguyen, Motohiro Sato, Ayako Honda, Stuart Gilmour.

**Visualization:** Phuong The Nguyen.

**Writing – original draft:** Phuong The Nguyen, Aminu Kende Abubakar, Phuong Mai Le.

**Writing – review & editing:** Phuong The Nguyen, Ryota Nakamura, Hideyasu Shimadzu, Aminu Kende Abubakar, Phuong Mai Le, Huy Van Nguyen, Hoa L. Nguyen, Motohiro Sato, Ayako Honda, Stuart Gilmour.

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
