## [Editor Report · Decision Letter 0]

Dear Dr Nguyen,

Thank you for submitting your manuscript entitled "Progress and inequality in child immunization in 38 African countries, 2000-2030: A spatio-temporal Bayesian analysis at national and sub-national levels" for consideration by PLOS Medicine.

Your manuscript has now been evaluated by the PLOS Medicine editorial staff and I am writing to let you know that we would like to send your submission out for external assessment.

However, we first need you to complete your submission by providing the metadata that are required for full assessment. To this end, please login to Editorial Manager where you will find the paper in the 'Submissions Needing Revisions' folder on your homepage. Please click 'Revise Submission' from the Action Links and complete all additional questions in the submission questionnaire.

Please re-submit your manuscript within two working days, i.e. by Mar 25 2025 11:59PM.

Once your full submission is complete, your paper will undergo a series of checks in preparation for external assessment

Kind regards,

Richard Turner PhD, for Suzanne De Bruijn, PhD

Consulting Editor, PLOS Medicine

plosmedicine@plos.org

---

## [Decision Letter · Decision Letter 1]

Dear Dr Nguyen,

Many thanks for submitting your manuscript "Progress and inequality in child immunization in 38 African countries, 2000-2030: A spatio-temporal Bayesian analysis at national and sub-national levels" (PMEDICINE-D-25-01067R1) to PLOS Medicine. The paper has been reviewed by subject experts and a statistician; their comments are included below and can also be accessed here: [LINK]

As you will see, the reviewers thought this was an interesting paper, but also had some concerns and requests for additional analyses. After discussing the paper with the editorial team and an academic editor with relevant expertise, I'm pleased to invite you to revise the paper in response to the reviewers' comments. We plan to send the revised paper to some or all of the original reviewers, and we cannot provide any guarantees at this stage regarding publication.

We ask that you submit your revision by May 21 2025 11:59PM. However, if this deadline is not feasible, please contact me by email, and we can discuss a suitable alternative.

Don't hesitate to contact me directly with any questions (sbruijn@plos.org).

Best regards,

Suzanne

Suzanne De Bruijn, PhD

Associate Editor

PLOS Medicine

sbruijn@plos.org

Comments from the academic editor:

Please revise to address the concerns of R2 and R3.

Comments from the reviewers:

Reviewer #1: Statistical review.

The authors explained their design well and gave good justifications for their model choices. INLA is an appropriate estimation method. The Bayesian models used are established techniques for spatial analysis and the Bayesian approach gave intuitive probability estimates. The full details on the models were in the supplementary. The authors included some residual checks, although some were on an absolute scale. There were extensive sensitivity analyses and additional details in the supplementary.

The main figures look good and are informative. They are relatively dense, but they give details on all countries. Close inspection shows some interesting differences, for example Chad and Burkina Faso, which are close on the socio-demographic index, but with very different predictions.

My only question is why was model bym_2 chosen when bym_1 had a smaller DIC and WAIC (Table S3)? Was the decision purely made on the absolute difference?

Minor comments

* Figure 5, try the ggrepel package to move the labels.

* Decimal places are not needed in Table S3 for the DIC and WAIC.

* Decimal places could be dropped in Tables S6 and S7. I think the number of countries and estimates makes the numbers hard to take in. Using integers would allow patterns to be discerned for a slight cost in accuracy. However, perhaps the point of these tables is for countries to use the data, in which case a higher accuracy is worthwhile.

* The authors could consider providing the maps and tables online in an interactive display. That could be useful for policy makers and staff in the individual countries.

* I would encourage the authors to share their R code, for example on GitHub. This increases transparency and allows others to build on their work.

Adrian Barnett

Reviewer #2: Overall, this is a well written and interesting paper, that makes important contribution to the literature. While there are previous studies discussing progress in vaccine coverage and inequalities, this paper discusses the feasibility of achieving basic immunization targets in Africa by 2030, which is informative and can also serve policymakers. I have some concerns and comments, which are outlined below.

Abstract:

"Childhood immunization coverage improved significantly across most of the African countries in our sample from 2000 to 2030, yet 13 countries were projected to fall short of full immunization at the national level." - This sentence can be split into two - one describing the past trend and one the projected change to improve clarity.

"Despite being high-SDI nations…" - It is unclear in the abstract what SDI stands for. Define the term.

Main text:

When the authors talk about the results of future projections they use past tense, for example, "Socioeconomic inequalities were widespread in 2000 but were

projected to decline or level off in 32 countries by 2030…". When referring to projected data, I would suggest avoiding past language, which sounds "factual", whereas these figures are based on assumptions about future trends. Generally, the authors could discuss the projected results with more caution and avoid "factual" language for example "X% of children are estimated to be vaccinated if the current trends continue…" or "countries are projected to reach X% vaccine coverage if they continue with the current rate of vaccination" or something like that.

The authors mention a target by 2030 of at least 80% coverage for essential vaccines, however, my understanding is that the target is 90%. See for example: https://www.who.int/teams/immunization-vaccines-and-biologicals/strategies/ia2030/explaining-the-immunization-agenda-2030?utm_source=chatgpt.com

The authors should correct the interpretation of the data accordingly, with potentially more countries set to miss the target.

In Figure 1, the countries could be ordered from best to worst performing, it would be easier to immediately identify those countries that are on course to achieve the target and those least likely to achieve it. The SDI ranking does not seem to add much to the figure. Also, in the caption of the figure, the authors should specify that the 2030 values are based on current trends or somehow indicate the implicit assumptions about the future course of action made in the 2030 figures.

Also in Figure 1, the authors show the AARC values from 2000 to 2030. I am not sure what the benefit of this AARC number is, since it includes both observed (and modeled) and projected data (including multiple assumptions about the future). Instead, the authors could present AARC values based on the long-term trend (2000 to 2020), and also based on a shorter-term trend (which is now included in the supplement) to show whether progress is accelerating or slowing down, this would add some interesting information (are we still on course or are we falling behind lately).

And then it could be interesting to calculate the minimum AARC needed to reach the 2030 target to demonstrate how much faster countries would need to increase vaccination efforts to catch up.

The authors could also try a few different assumptions for the future, for example vaccination levels in 2030 if AARCs were same, if they were based on the best performing region in each country, or something like that. The authors could play around with these assumptions a bit more to demonstrate different future scenarios depending on the course taken. Just as an idea.

How do the authors deal with countries when they only have 1 survey for that country? This is the case with Angola, Comoros, Cote d'Ivoire, Eswatini, and a few other countries. How are values for the period 2000-2020 and AARCs calculated in that case? The authors should be more explicit about their assumptions and modeling.

When calculating the SII and RII for 2030, did the authors use rates of change specific to each wealth/education group? Or did they apply the country-level AARCs?

"By 2030, wealth-related inequalities were projected to decline in many subnational

regions." Again, this result is driven by the assumptions made about the future trends. What were these assumptions, what were the rates assumed for each wealth/education group?

Concerning the projections, again it may be interesting to add what AARCs are needed for each country/region to reach the 2030 goals and also to close the socioeconomic gaps.

In Figure 5, it may be better to use the data for 2000 and 2020, instead of 2000 and 2030, to present factual (modeled?) progress. More discussion is needed about the difference in the interpretation of SII and RII, what do each of the groups mean? What does it mean about where progress needs to be made? What is the benefit of this figure and the presented information?

Reviewer #3: Overview:

In this manuscript, the authors use spatiotemporal modeling approaches (BYM) to estimate routine childhood immunization coverage for five indicators stratified by space, time, and socioeconomic status, while also projecting trends through 2030 and conducting inequality analyses. In general, the manuscript is very well written; the analysis benefits from a large data set as well as a thorough consideration of both coverage patterns and inequality across multiple dimensions. My comments below mainly focusing on the technical aspects of the manuscript and methods, as the results and discussion are quite thorough.

Major comments:

1. Introduction: I would suggest that the authors more fully review the literature and cite some of the previous multi-country modelling studies of subnational immunization coverage that have been conducted at regional and continental levels. This study does provide substantial novelty beyond previous efforts to estimate subnational coverage, including via stratification by socioeconomic status and producing projections, but a clearer description of prior work in this area and comparison / contrast to the present work would strengthen the introduction.

2. Methods: The authors use DHS surveys; did they also consider using other surveys that report subnational immunization coverage data (MICS etc.?) Some discussion of this would be helpful, even if in the limitations section.

3. Methods: do the authors implement any constraints on coverage, particularly for full immunization coverage (ie, FIC has to be less than each of BCG/Polio3/DPT3/MCV1 coverage?). If not, it would be useful to know whether there are locations where the model predicts higher FIC coverage than coverage of one of the other vaccines; ie, to what extent are these estimates internally consistent?

4. Methods: Why were studies only included if conducted between 2000-2019 (that is, why were more recent studies excluded?) Particularly given the impact of the COVID-19 pandemic on childhood immunization coverage, it would seem like it would be particularly important to include what data is available from the last 5 years if at all possible, as I might expect that this would affect current estimates and particularly future projections. I appreciate that this is given as a limitation in the discussion, but would be interested to understand why the data weren't included instead - as this does seem a rather substantial limitation.

5. Supplemental methods / spatio-temporal models: If I understand the model correctly, the authors model a global temporal linear trend (beta) in addition to individual areal-unit-level effects. One might expect that the overall trends in vaccine coverage are likely to vary within countries substantially - did the authors consider a model wherein country-level aggregate time trends were estimated rather than a single global time trend across all countries, ie. Beta_c where c varies by country? Or are there technical reasons why such a model would not be possible to fit? The same holds true for the BYM2 model, for instance, described on page 7, which also uses a single global time trend, and I think also to essentially all of the models 1-5 which also include beta_t.

6. Methods / Supplemental methods: Related to the above, the authors note that the various model formulations are essentially assuming a linear time trend for each area (or, perhaps more precisely, that logit(coverage) is linearly related to time). If I follow the model specifications correctly, in locations with a single survey the temporal trends are essentially driven by the global time trend (beta_t). This is a relatively major assumption and can lead to some dramatic time trends.

a. Did the authors consider alternative methods of attempting to capture trends in time (e.g. using autoregressive or random walk models for the temporal component rather than a linear model, and/or including interactions between the spatial and temporal terms? Or alternatively using covariates to help in form the temporal trends (though appreciate that this may complicate forecasting)?

b. For instance, looking at the model_bym_2 results (Figure S8), BCG coverage increases dramatically in Angola from 2000-2020, by more than 25 percentage points in many locations. If we look at the WUENIC estimates for Angola, however, which include multiple survey time points and incorporate administrative data trends, BCG coverage is essentially estimated to be flat over this time period - with some increases and decreases, but much different than the dramatic continuous increase suggested by this model. I give Angola as one example, but in many cases the time trends implied here are quite different than what are implied by WUENIC. I bring this up mainly because the effects on projection are likely to be substantial - looking at the WUENIC estimates, one wouldn't necessarily expect coverage to increase dramatically by 2030. I appreciate that these are very different estimation methodologies, but it would be helpful to understand how these estimates (aggregated to the national level) compare to the widely-used WUENIC estimates.

7. Methods: the authors write that they "conducted stratified analyses by subgroups and incorporated random slopes to account for variations in these variables across locations and included interaction terms between these variables and survey year". The formulas in the supplementary materials, however, seem to be only for the more general spatio-temporal models, and I don't see anything in the descriptions or equations / notations that specifies how this was done. Some additional detail describing this process and perhaps showing these results would be helpful as the stratified analysis inequality is one of the more novel parts of this manuscript, and without additional description of how this was implemented it's hard for the reader to fully interpret the results. For instance, did the authors run a true stratified analysis with separate models for each of the subgroups? Or did they run a single model that incorporated both random slopes by location (were these hierarchical in any way?) and interactions by survey year? If this is the case, then these seem like they could be highly flexible models - including random slopes on the covariates that vary by subnational location and over time, while also allowing for the spatial effects in the BYM model. I would think that the model might face challenges in accurately estimating all of these location-specific effects, particularly for subnational regions that only have a single survey-year of data if so, but perhaps that wasn't the authors' experience. In any case, a more detailed description of the methods used to accomplish this would be very helpful in the supplemental materials and allow for a more thorough understanding of the results presented here.

8. Methods: The authors validate the model using average absolute difference. A few questions about model validation:

a. It might be helpful to also give some additional measures, like the mean error (to assess whether the model is systematically biased either high or low, which metrics like the average absolute difference can't tell us) and the 95% coverage of the uncertainty or predictive intervals (to assess how well uncertainty is calibrated in the model)

b. I would encourage the authors to also provide some out-of-sample validation statistics for their models, as all of the validation metrics here appear to be in-sample if I am reading them correctly. This could be accomplished by holding out surveys, for instance, in a cross-validation framework. As the authors are also projecting into the future, it would be helpful to see how accurately the model is able to make forward projections, so some validation by withholding recent years of data and seeing how well the model can predict based on past trends would be very useful.

9. Results: The authors write that "The sensitivity analysis using recent trends (2010-2020), presented in Supplementary Figure S27, shows that these trends generally project lower Full immunization coverage for 2030 and slower progress with smaller AARC values. In addition, recent trends appear less effective in capturing regional disparities and variations, likely due to data limitations. In contrast, overall trends (2000-2020) provide more robust projections, with higher coverage levels and better representation of regional diversity". An alternative explanation may be that coverage gains have slowed or plateaued in many locations in more recent years, which has been a phenomenon observed at global, regional, and select national levels by WUENIC among other sources. The maps in S23 do appear to demonstrate, as might be expected, that data sparsity in the model fit using only 2010-2020 shows less regional variation simply due to the exclusion of many previous surveys. I would be cautious, however, with the assertion that overall trends using 2000-2020 provide more robust estimates simply because they are higher in coverage, as one could argue that the inclusion of the earlier years may instead be overestimating the expected gains between 2020-2030, particularly given the relative simplicity of the projection method used here (a continuation of the logit-linear relationship between coverage and time). The authors explain this well in the subsequent discussion, but I think that the framing in the results section might benefit from some of the same nuance.

10. General: While the manuscript focuses on Africa, the authors' affiliations appear to be all from institutions outside of Africa. I would suggest that the authors seek to include the perspectives of colleagues and experts from the countries included in this study to inform and contextualize these findings if not already done, while providing appropriate acknowledgements and/or authorship to these individuals depending on their contributions.

Minor comments:

1. Please add line numbers to subsequent revisions of the manuscript to facilitate reviewer comments if possible

2. Methods and throughout: I would suggest specifying in the methods that measles here is presumably MCV1, and not the second dose of measles (since Polio3 and DPT3 have the dose number indicated in the indicator definition).

3. Methods: it would be helpful to have some additional information on how the data were geolocated and what shapefile was used for the analysis. From looking at the maps it appears that the unit of analysis was generally the first administrative level; were the data matched to shapefiles from DHS? A few lines explaining explicitly what the subnational unit of analysis was and what shapefile was used would be helpful. Were there any instances in which subnational boundaries changed over the study period; if so, would also explain how those were handled?

4. For all figures, I would consider providing at least one additional year that indicates the estimates at the end of the modeling period, to help distinguish somewhat between the estimates in years for which data was available and those which were purely projections. For instance, it would be useful in the maps in Figure 2 and Figure 4 to see not only the year 2000 and the year 2030, but also the last year in which the authors have available data (appears to be 2019)? Figure 1 shows this nicely by including 2000, 2020, and 2030, so that also could be an alternative approach. The 30 year gap between 2000 and 2030 hides some of the benefit of the analysis in producing a quasi-contemporaneous set of estimates for the present.

5. For figures S5-S9, is there a reason why these are given for BCG only? It would be helpful to see these for the other indicators as well for completeness. Since the authors are providing results through 2030, I would also suggest extending these plots to include the projections through that year.

6. Last paragraph of results: "The sensitivity analysis using recent trends (2010-2020), presented in Supplementary Figure S27" - I think that this is S23? I would double-check the numbering of all figures to ensure that there wasn't an alteration that caused these to be mismatched.

---

* Please upload any figures associated with your paper as individual TIF or EPS files with 300dpi resolution at resubmission; please read our figure guidelines for more information on our requirements: http://journals.plos.org/plosmedicine/s/figures. While revising your submission, please upload your figure files to the PACE digital diagnostic tool, https://pacev2.apexcovantage.com/. PACE helps ensure that figures meet PLOS requirements. To use PACE, you must first register as a user. Then, login and navigate to the UPLOAD tab, where you will find detailed instructions on how to use the tool. If you encounter any issues or have any questions when using PACE, please email us at PLOSMedicine@plos.org.

* Please ensure that the study is reported according to the RECORD guideline and include the completed RECORD checklist as Supporting Information. When completing the checklist, please use section and paragraph numbers, rather than page numbers. Please add the following statement, or similar, to the Methods: "This study is reported as per RECORD guideline (S1 Checklist)."

FIGURES AND TABLES

SUPPLEMENTARY MATERIAL

REFERENCES

* Please ensure that the study is reported according to the RECORD guideline (available from https://www.record-statement.org) and include the completed checklist as Supporting Information. Please add the following statement, or similar, to the Methods: "This study is reported as per the Reporting of Studies Conducted using Observational Routinely-Collected Data (RECORD) guideline (S1 Checklist)." When completing the checklist, please use section and paragraph numbers, rather than page numbers.

* For all observational studies, in the manuscript text, please indicate: (1) the specific hypotheses you intended to test, (2) the analytical methods by which you planned to test them, (3) the analyses you actually performed, and (4) when reported analyses differ from those that were planned, transparent explanations for differences that affect the reliability of the study's results. If a reported analysis was performed based on an interesting but unanticipated pattern in the data, please be clear that the analysis was data driven.

* Please state in the Methods section whether the study had a prospective protocol or analysis plan. If a prospective analysis plan (from your funding proposal, IRB or other ethics committee submission, study protocol, or other planning document written before analyzing the data) was used in designing the study, please include the relevant document(s) with your revised manuscript as a Supporting Information file to be published alongside your study and cite it in the Methods section. A legend for this file should be included at the end of your manuscript. If no such document exists, please make sure that the Methods section transparently describes when analyses were planned, and when/why any data-driven changes to analyses took place. Changes in the analysis, including those made in response to peer review comments, should be identified as such in the Methods section of the paper, with rationale.

The following list is derived from Geoffrey P Garnett, Simon Cousens, Timothy B Hallett, Richard Steketee, Neff Walker. Mathematical models in the evaluation of health programmes. (2011) Lancet DOI:10.1016/S0140-6736(10)61505-X:

* If pertinent, please provide a diagram that shows the model structure, including how the natural history of the disease is represented, the process and determinants of disease acquisition, and how the putative intervention could affect the system.

* Please provide a complete list of model parameters, including clear and precise descriptions of the meaning of each parameter, together with the values or ranges for each, with justification or the primary source cited and important caveats about the use of these values noted.

* Please provide a clear statement about how the model was fitted to the data, including goodness-of-fit measure, the numerical algorithm used, which parameter varied, constraints imposed on parameter values, and starting conditions.

* For uncertainty analyses, please state the sources of uncertainties quantified and not quantified [can include parameter, data, and model structure].

* Please provide sensitivity analyses to identify which parameter values are most important in the model. Uncertainty estimates seek to derive a range of credible results on the basis of an exploration of the range of reasonable parameter values. The choice of method should be presented and justified.

* Please discuss the scientific rationale for the choice of model structure and identify points where this choice could influence conclusions drawn. Please also describe the strength of the scientific basis underlying the key model assumptions.

* For studies that develop a prediction model or evaluate its performance, please ensure that the study is reported according to the TRIPOD statement (https://www.equator-network.org/reporting-guidelines/tripod-statement) and include the completed checklist as Supporting Information. Please add the following statement, or similar, to the Methods: "This study is reported as per the Transparent Reporting of a Multivariable Prediction Model for Individual Prognosis Or Diagnosis (TRIPOD) statement (S1 Checklist)." For studies using machine learning, please use the TRIPOD-AI checklist. When completing the checklist, please use section and paragraph numbers, rather than page numbers.

---

## [Decision Letter · Decision Letter 2]

Dear Dr. Nguyen,

Thank you very much for re-submitting your manuscript "Progress and inequality in child immunization in 38 African countries, 2000-2030: A spatio-temporal Bayesian analysis at national and sub-national levels" (PMEDICINE-D-25-01067R2) for review by PLOS Medicine.

I have discussed the paper with my colleagues and the academic editor and it was also seen again by 2 reviewers. I am pleased to say that provided the remaining editorial and production issues are dealt with we are planning to accept the paper for publication in the journal.

We have the following editorial comments, based on the reviewer comments:

1) We agree that is makes sense to keep the SDI ordering in Figure 1.

2) Could you modify figure 1 to make the numbers in the columns more readable, as commented on by R1? we are happy to keep the 3 columns, but would prefer the numbers to be easier to read.

3) Please address the remaining two concerns from R3 with textual changes

4) Please address the comment of R1 regarding the clarity in line 355.

5) Please consider modifying figure 5 to make the datapoints easier to distinguish.

6) Regarding the R code, Please provide us with a link to the Github repository. Ideally, we would like you to provide us with a DOI (e.g. generated by Zenodo).

In addition to these comments, we have several editorial comments, that can be found at the bottom of this email. Please address all these issues, and provide us with a reply to these comments.

Any accompanying reviewer attachments can be seen via the link below. Please take these into account before resubmitting your manuscript:

[LINK]

We look forward to receiving the revised manuscript by Jun 26 2025 11:59PM.   

Sincerely,

Suzanne De Bruijn, PhD

Associate Editor 

PLOS Medicine

plosmedicine.org

Requests from Editors:

* Please confirm that your title complies with to PLOS Medicine's style. Your title must be nondeclarative and not a question. It should begin with main concept if possible. "Effect of" should be used only if causality can be inferred, i.e., for an RCT. Please place the study design ("A randomized controlled trial," "A retrospective study," "A modelling study," etc.) in the subtitle (ie, after a colon).

* Please confirm that your abstract complies with our requirements, including format (three sections: Background, Methods and Findings, and Conclusions) and providing all the information relevant to this study type https://journals.plos.org/plosmedicine/s/submission-guidelines#loc-abstract

* At this stage, we ask that you include a short, non-technical Author Summary of your research to make findings accessible to a wide audience that includes both scientists and non-scientists. The Author Summary should immediately follow the Abstract in your revised manuscript. This text is subject to editorial change and should be distinct from the scientific abstract. Ideally each sub-heading should contain 2-3 single sentence, concise bullet points containing the most salient points from your study. In the final bullet point of ‘What Do These Findings Mean?’ Please include the main limitations of the study in non-technical language.

Please see our author guidelines for more information: https://journals.plos.org/plosmedicine/s/revising-your-manuscript#loc-author-summary.

* Please check that any use of statistical terms (such as trend or significant) are supported by the data, and if not please remove them.

* Please remove the 'conclusions' subheading from the discussion. Please also remove any other subheadings from the discussion.

* Please consider avoiding the use of red and green in order to make your figure more accessible.

* Please confirm that the appropriate usage rights apply to the use of the maps. Please see our guidelines for map images: https://journals.plos.org/plosmedicine/s/figures#loc-maps

* PLOS has a 'Inclusivity in Global Research' policy which aims to promote collaboration and inclusivity in global health research. You are required to complete PLOS’ questionnaire on inclusivity in global research and submit it with your revised paper. The policy and questionnaire can be found at https://journals.plos.org/plosone/s/best-practices-in-research-reporting.

*As you state no ethical approval is needed, please amend your ethics statement.

* Please ensure that the study is reported according to the RECORD guideline and include the completed RECORD checklist as Supporting Information. When completing the checklist, please use section and paragraph numbers, rather than page numbers. Please add the following statement, or similar, to the Methods: "This study is reported as per RECORD guideline (S1 Checklist).

*please modify the following sentence to ensure correct citation formatting, by removing the years: "“Notably, prior multi-country modelling efforts, such as Mosser et al. (2019) and Utazi et al. (2018), have mapped fine-scale immunization coverage across Africa, providing valuable spatial insights, but they did not incorporate socioeconomic stratifications, forward projections, or detailed inequality analyses at sub-national levels [17, 18].” You could consider "modelling efforts, e.g. [8, 9]", or "shown in the modelling work by Mosser and colleagues [17] and Utazi et al. [18]".

Comments from Reviewers:

Reviewer #1: The methods and data appear sound. I can imagine the results being useful for policy in many African countries. The paper comes at an important time given the recent disruption to aid programs.

I have only minor comments which are entirely optional for the authors to change or not.

Figure 1 now includes a third column of numbers, and the numbers are now a little squeezed in. For example, some of the negative signs for AARC are barely visible (e.g., Egypt, full immunisation) and some are potentially cut as there's no room. I prefer the previous version with two rows of numbers as I think this new version is too cluttered.

The authors could consider hexagonal maps in future research as an alternative to standard choropleth maps (e.g., Figure 2), see the sugarbag package in R https://github.com/srkobakian/sugarbag. This approach will better show the geographically smaller countries.

Line 335: "This pattern of disparity was consistent across all immunization indicators, although BCG and MCV1 had higher coverage than DPT3 and Polio3". Could be more specific about "higher" by giving an average percentage.

Some white space in Figure 5 could be reduced by reducing the upper limits of the x and y axis to 1.5 and 4, respectively. This would allow the estimates of more countries to be clearly seen. You could consider drawing a circle centred on the (0,0) coordinate with a suitable radius that indicated no meaningful change.

Figure 6. "m" in the facet headings is potentially vague and could be changed to "months", e.g., "12 to 23 months".

Line 607. Providing the link to the Github repository would be useful.

Reviewer #3: Many thanks to the authors for all of their responses and additions to the manuscript. My questions have all been thoroughly addressed. My only remaining comments (see below) are on major comment #6, regarding the characterization of the WUENIC estimates, and major comment #8, where I think that a sentence reflecting on the mean error out of sample validation results would be a helpful addition. Otherwise, I thank the authors for their thorough work and believe that the modifications made in response to reviewers' comments have strengthened the manuscript.

Previous major comments

1. Regarding the review of the literature - I appreciate the additional sentences added, which help to contextualize this work. No additional comments on this topic.

2. Regarding the use of non-MICS surveys - well addressed; no further comments.

3. Regarding constraints on FIC < individual vaccine coverage - I appreciate the added text in the limitations; well addressed and no further comments.

4. Regarding the lack of surveys after 2019 - the author's justification is well taken and I appreciate the added text in the limitations. I particularly appreciate the current challenges in accessing additional DHS data given the termination of the DHS program. No further comments.

5. Regarding the modeling of time trends - I appreciate the author's clarification, and the revised methods section is clearer to the reader. Many thanks for the explanation; no further questions

6. Regarding the use of the logit-linear model for time trends: I appreciate the author's explanation and the additional text added to the analysis and limitations section. Within the constraints of this temporal modeling approach, I think that the text is now clear and appropriately conveys the limitations. My one additional comment is that WUENIC's estimates are not technically "administrative" estimates, but instead use a rule-based approach to integrate both survey and administrative (or official country-reported) data, so the language here may need to be slightly revised.

7. Regarding the stratified analyses: I appreciate the authors' explanation, and the additions to the supplementary methods are extremely useful in clarifying how the model was set up. No further questions.

8. Regarding model validation: I appreciate the addition of these analyses - both the out of sample validation and the addition of mean error. My one remaining comment is that it might be useful to add a brief sentence about mean error to the main text. In lines 255-263 the authors comment on the average AD and ranges; a similar statement & interpretation of the mean error results would be useful to the reader. In particular it is interesting that the in-sample mean error is essentially zero, while the out-of-sample mean error (holding out 2017-2019) is consistently positive, ranging from 1.27% to 7.86% for the BYM2_2 model across indicators. This might suggest a tendency towards overestimation in more recent years in the absence of data, which is worth pointing out as a potential limitation (perhaps reflecting some of the additional discussion about coverage stagnation and the limitations of forward-projection using the logit-linear time trend approach). A sentence in the main text reflecting these findings would be helpful for readers' interpretation of the results.

9. Results: regarding the potential role of stagnation in coverage in the results presented here - the authors' explanation is clear and well received, and the edits to the text are excellent. I particularly appreciate the addition of the sensitivity analysis using 2010-2019 which is a good addition to the paper and provides some additional detail that is useful to the readers and suggests slowdowns in progress.

10. Regarding the inclusion of perspectives from colleagues in Africa: I appreciate the authors' response and the acknowledgements provided here; no further comments in this regard.

Minor comments:

1. Addressed

2. Addressed

3. Many thanks to the authors for the additional methodological description, which are excellent. Fully addressed.

4. Well addressed; many thanks to the authors for these extensive additions. No further comments.

5. Fully addressed; thanks to the authors. I agree that the approach used here does a good job of providing additional information without overwhelming the reader.

6. Addressed

[LINK]

---

## [Editor Report · Decision Letter 3]

Dear Dr Nguyen, 

On behalf of my colleagues and the Academic Editor, Rebecca Grais, I am pleased to inform you that we have agreed to publish your manuscript "Progress and inequality in child immunization in 38 African countries, 2000-2030: A spatio-temporal Bayesian analysis at national and sub-national levels" (PMEDICINE-D-25-01067R3) in PLOS Medicine.

Before your manuscript can be formally accepted, we have two additional editorial requests:

1) Thanks for including the RECORD checklist. However, I noted the page numbers are incorrect. Please amend, and include paragraph numbers.

2) Please remove the funding statement from the manuscript.

PRESS

Sincerely, 

Suzanne De Bruijn, PhD 

Associate Editor 

PLOS Medicine